# Replication of Sensor-Based Categorization of Upper-Limb Performance in Daily Life in People Post Stroke and Generalizability to Other Populations

**DOI:** 10.3390/s25154618

**Published:** 2025-07-25

**Authors:** Chelsea E. Macpherson, Marghuretta D. Bland, Christine Gordon, Allison E. Miller, Caitlin Newman, Carey L. Holleran, Christopher J. Dy, Lindsay Peterson, Keith R. Lohse, Catherine E. Lang

**Affiliations:** 1Program in Physical Therapy, Washington University School of Medicine, St. Louis, MO 63110, USA; mchelsea@wustl.edu (C.E.M.); blandm@wustl.edu (M.D.B.); gordon.christine@wustl.edu (C.G.); miller.allison@wustl.edu (A.E.M.); cholleran@wustl.edu (C.L.H.); lohse@wustl.edu (K.R.L.); 2Department of Neurology, Washington University School of Medicine, St. Louis, MO 63110, USA; 3Program in Occupational Therapy, Washington University School of Medicine, St. Louis, MO 63110, USA; 4Shirley Ryan AbilityLab, Chicago, IL 60611, USA; cnewman@sralab.org; 5Department of Orthopedic Surgery, Washington University School of Medicine, St. Louis, MO 63110, USA; dyc@wustl.edu; 6Department of Medicine, Washington University School of Medicine, St. Louis, MO 63110, USA; llpeterson@wustl.edu

**Keywords:** activities of daily living, measurement, musculoskeletal, neurology, rehabilitation, upper limb, wearable sensors

## Abstract

**Highlights:**

**What are the main findings?**
A five-variable, five-cluster model was replicated in people with stroke and controls, and it generalized to musculoskeletal and other neurological conditions affecting the upper limb.Compared to clusters, two principal components and individual accelerometry variables showed higher convergent validity with self-report outcomes of upper limb performance and disability.

**What is the implication of the main finding?**
Upper limb performance in daily life, quantified by wearable movement sensors, may be better represented on a continuum of functional recovery, rather than with discrete categories.This application of wearable movement sensors supports a unified, data-driven approach to monitor upper limb recovery across conditions and severity of functional deficits in rehabilitation.

**Abstract:**

Background: Wearable movement sensors can measure upper limb (UL) activity, but single variables may not capture the full picture. This study aimed to replicate prior work identifying five multivariate categories of UL activity performance in people with stroke and controls and expand those findings to other UL conditions. Methods: Demographic, self-report, and wearable sensor-based UL activity performance variables were collected from 324 participants (stroke *n* = 49, multiple sclerosis *n* = 19, distal UL fracture *n* = 40, proximal UL pain *n* = 55, post-breast cancer *n* = 23, control *n* = 138). Principal component (PC) analyses (12, 9, 7, or 5 accelerometry input variables) were followed by cluster analyses and numerous assessments of model fit across multiple subsets of the total sample. Results: Two PCs explained 70–90% variance: PC1 (overall UL activity performance) and PC2 (preferred-limb use). A five-variable, five-cluster model was optimal across samples. In comparison to clusters, two PCs and individual accelerometry variables showed higher convergent validity with self-report outcomes of UL activity performance and disability. Conclusions: A five-variable, five-cluster model was replicable and generalizable. Convergent validity data suggest that UL activity performance in daily life may be better conceptualized on a continuum, rather than categorically. These findings highlight a unified, data-driven approach to tracking functional changes across UL conditions and severity of functional deficits.

## 1. Introduction

Upper limb (UL) use is integral for engagement in activities of daily life. When one or both ULs are affected due to conditions such as stroke or shoulder pain, the ability to perform daily activities may become challenging. This can substantially disrupt one’s functional independence and participation, leading to disability [1]. People often seek out rehabilitation services to improve their ability to perform activities in daily life [2]. In-clinic measurement tools accurately quantify a person’s capacity for UL activities within a clinical setting (i.e., activity capacity), but these measures do not quantify what a person actually does in an unstructured, free-living environment (i.e., activity performance in daily life) [3,4]. In recent years, wearable movement sensors such as accelerometers have emerged as a research tool to quantify UL activity performance in daily life and their potential for integration in clinical rehabilitation continues to grow [5,6].

Rehabilitation clinicians manage diverse patient populations and require measurement tools that are efficient, easily understood by themselves and their patients, and adaptable to the time constraints of high-demand care environments. As wearable movement sensors transition into the clinical space, there are multiple challenges that need to be addressed: (1) the number of single variables used across studies, often with different definitions or algorithms, creates inconsistency in the description of UL activity performance in daily life; (2) the mathematical complexity and interpretability of some variables complicates their clinical use; (3) the lack of validation data hinders understanding of how UL accelerometry variables may relate to self-reported UL use in daily life; and (4) the generalizability across various clinical populations and severity of functional deficits remains unclear [7,8]. A potential solution to overcome some of these challenges would be to transition from using multiple single variables that describe unique populations to multivariate categories of UL activity performance in daily life that could span clinical populations.

Several single sensor variables have been validated to capture UL activity performance in daily life in people post stroke [8]. While single variables may be highly useful, UL activity performance in daily life is likely multi-dimensional [9]. Thus, quantifying UL activity performance with single variables may not fully represent UL activity performance in daily life [9,10]. In prior work, we identified five distinct multivariate categories that characterized UL activity performance in daily life in a sample of people with stroke and people without UL disability. Using different numbers of accelerometry input variables, two principal components consistently accounted for the majority of the variance, and a five-cluster solution provided the best model fit by maximizing the overall variance explained [11]. Although these results were promising, additional studies are needed to replicate and validate the findings. Further, if a solution for the clinical environment involves multivariate categories, then they should generalize beyond stroke to other conditions affecting one or both ULs, thus improving clinical utility through broader applicability [7,8].

This manuscript is organized according to the primary purposes of this study, which were to (1) replicate the multivariate categories of UL activity performance in a separate sample of people with stroke and people without UL disability; (2) determine the generalizability of the categories beyond stroke to other conditions for which people seek out UL rehabilitation; and (3) evaluate the convergent validity between sensor-based categories and self-reported measures of daily UL activity and quality of life. To enhance generalization, we aimed to include a heterogeneous sample of clinical populations known to cause UL disability who seek rehabilitation services. Since it would be impossible to study every condition, we intentionally selected candidate conditions that varied in their biological underpinnings (i.e., neurological: stroke, multiple sclerosis; musculoskeletal: shoulder pain, fractures; and medical: post curative breast cancer treatment), region of impact (e.g., proximal, distal, or entire limb), and range of functional severity and chronicity (e.g., distal radius fracture, multiple sclerosis). We hypothesized that five multivariate categories would be replicable in a new sample of people with stroke and people without UL disability and that these categories would be generalizable across conditions and severity of functional deficits despite variable biological causes and patterns of UL impairment and capacity. We further hypothesized that convergent validity of the categories would be evident between condition-specific and generic self-report measures of UL activity performance and disability. If confirmed, defined categories from wearable movement sensor data that can be used across clinical populations and functional levels of severity could dually promote personalized care and clinical efficiency within rehabilitation settings.

## 2. Materials and Methods

This study enrolled two groups of people into a prospective, longitudinal, observational cohort: (1) adults referred to physical or occupational therapy services for UL problems and (2) adults who had no history of neurological or musculoskeletal conditions that affected their ULs to serve as controls. The data used in this report were from the first assessment, which occurred within 2 weeks of starting rehabilitation care for those receiving services. By nature of its design, this study could either be administered in-person during a clinic visit or fully remotely depending on participant preference. This study used a single Institutional Review Board (IRB) at Washington University (WashU IRB# 202207003-1001, approval date 21 July 2022). All participants provided informed consent, with most consenting electronically via REDCap version 15 [12,13,14,15].

### 2.1. Participants

Participants included people seeking rehabilitation services for conditions affecting the UL (e.g., stroke, multiple sclerosis, UL fracture, shoulder pain, breast cancer) and people without UL disability as comparators. Participants with conditions affecting the UL were recruited from rehabilitation and medical clinics at WashU Medicine in St. Louis, Missouri, and Shirley Ryan Ability Lab in Chicago, Illinois, or from outside either medical network across the United States via electronic flyers, condition-specific websites, and social media advertisements. An additional source to recruit control participants was the Recruitment Enhancement Core through the Institute of Clinical and Translational Sciences services at Washington University.

#### 2.1.1. Participants with Conditions Affecting the Upper Limb


*General inclusion criteria:*
Age > 18 years.UL disability determined by the referring physician or surgeon.Referred to rehabilitation to address UL disability.Documented goals of service (rehabilitation and/or surgery) to increase or restore UL function.



*Condition specific inclusion criteria:*
Stroke: Confirmed ischemic or hemorrhagic stroke diagnosis by neurologist, consistent with imaging; unilateral UL sensorimotor impairments due to stroke.Multiple sclerosis (MS): Confirmed diagnosis of MS by neurologist; sensorimotor impairments in at least one UL.Distal UL fracture: Unilateral, radiographically confirmed, distal radius fracture, either treated with surgery or non-operatively.Proximal UL Pain: Unilateral, radiographically confirmed, proximal humerus or clavicle fracture, either treated with surgery or non-operatively or physician diagnosis of shoulder pain of musculoskeletal origin; limitations in shoulder range of motion; and reported problems using the limb for functional activities.Breast Cancer: Confirmed diagnosis of breast cancer, stage 0–III, by oncology provider; >4 weeks post curative-intent breast cancer treatment (treatment could include one or more of surgery, chemotherapy, and radiation). This could be a new or older diagnosis of breast cancer. Participants could have unilateral or bilateral UL involvement and were not excluded if lymphedema was present.



*General exclusion criteria for participants with UL conditions:*
Other concurrent neurologic, musculoskeletal, or medical conditions that affected the UL (e.g., exclude if both stroke plus distal radius fracture) or general physical activity.Other co-morbid conditions that indicate a minimal chance for functional improvement (e.g., end-stage cancer, end-stage renal disease).Pregnant or planning to become pregnant.Cognitive impairment or disorders of communication that would prevent informed consent and study completion as indicated in their medical record.


#### 2.1.2. Participants Without UL Disability Serving as Controls


*Eligibility criteria:*
Age > 18 years.No neurological, musculoskeletal, or medical conditions that affect the UL, or that significantly affect the ability to engage in physical activity as reported by the participant.


Once enrolled, participants chose the most feasible method of study participation. Depending on participant choice, the described data collection and assessment procedures could occur (1) remotely via mailing of sensors, electronic questionnaires (via REDCap version 15), and secure telephone or zoom calls; (2) during an in-person home visit; or (3) during an in-person clinic visit. Study participants could also opt to have some aspects administered remotely with others administered in person.

### 2.2. Study Assessments

UL activity performance in daily life was quantified directly from bilateral wrist worn accelerometers and indirectly from self-report. Descriptive demographic and quality of life data were collected via self-report or via the electronic health record.

#### 2.2.1. Accelerometry Measurement of Upper Limb Performance in Daily Life

The devices used were the tri-axial ActiGraph GT9x-BT Links. Ametris (formerly ActiGraph LLC, Pensacola, FL, USA) is a Class II FDA-Approved Medical Device which has established reliability and validity standards and conforms with requirements of the International Standardization Organization [16] to meet regulatory requirements and ensure high quality medical devices. Participants were instructed to continuously wear the accelerometers on both wrists, with wrist straps comfortably taught, and positioned just proximal to the ulnar styloid for two consecutive days [17]. Past literature has indicated that a minimum of 24 h is sufficient to show stability of UL activity performance variables in adults [17,18] and that shorter durations of prescribed wear time often indicate greater adherence [19]. Participants were also asked to keep a log of wear time, rate their activity level over the course of the two days, document any times the sensors were removed, and report any difficulty wearing the devices. After two days, participants returned the accelerometers to their research facility (in person, by mail).

Once accelerometers were returned, recorded data were downloaded using ActiLife 6 (ActiGraph LLC, Pensacola, FL, USA), visually inspected, and processed using custom code programmed in R software version 4.4.2 (R Core Team, Vienna, Austria) [11,17,20]. The processing code can be found on the following Zenodo repository: https://doi.org/10.5281/zenodo.10999195 [21,22]. For inclusion in analyses, participants had to have at least one valid day (≥24 h) on bilateral devices [18]. There were two data files extracted from ActiLife 6 software: a 1 Hz data file (activity counts) and a 30 Hz data file (gravitational units). Data processing methods were replicated from Barth et al., 2021, [11] where most variables were computed from the vector magnitude of the x, y, and z axes accelerations after bandpass filtering (between 0.25 and 2.5 Hz), converting to activity counts, and down sampling into 1-s epochs using proprietary ActiLife software [17,23]. A few variables (e.g., Preferred/Non-Preferred Spectral Arc Length, and Jerk Asymmetry Index) that required higher sampling frequencies were computed from 30 Hz data using the vector magnitude of the x, y, and z accelerations in gravitational units after bandpass filtering (0.2–12 Hz).

Twelve UL activity performance variables, identical to those described by Barth et al., 2021, were included in this analysis (Table 1) [11]. In contrast to Barth et al., the current report uses the terms preferred and non-preferred limbs for simplicity [11]. The preferred limb is the dominant limb of the control participants and the non-affected/non-injured limb of the patient participants. Likewise, the non-preferred limb is the non-dominant limb or the affected/injured limb, respectively.

#### 2.2.2. Self-Report Measurements of Upper Limb Performance in Daily Life

Most participants completed online questionnaires remotely via REDCap (Vanderbilt University, Nashville, TN, USA), with a few participants who completed them either on paper or on a computer with assistance from study personnel. All surveys were completed by participants with the exception of the Disability of the Shoulder Arm and Hand (DASH) survey, which was only completed by people with musculoskeletal conditions affecting the UL (e.g., breast cancer, proximal shoulder pain or fracture, and distal fracture groups), and the Motor Activity Log–Amount of Use Scale (MAL-AoU), which was only completed by people with neurological conditions affecting the UL (e.g., stroke and multiple sclerosis groups).

#### 2.2.3. Patient-Reported Outcome Measurement Information System Upper Extremity Bank 2.0 via Computer Adaptive Test (PROMIS)

The PROMIS is a system of self-reported measurement tools that evaluate health status for physical, mental, and social well-being. This study used the computer adaptive test for the UL as the primary measure to quantify participant perception of UL activity performance in daily life. The Upper Extremity Bank 2.0 consists of 126 items, with participants typically answering only 5–6 questions for the computerized adapted testing version. Scores are reported as T-scores, where the normative sample without any conditions has a mean ± SD of 50 ± 10 points [28]. PROMIS tools have excellent test–retest reliability and internal consistency [28,29,30], and they have been used across a wide variety of chronic diseases and conditions and in the general population [29].

#### 2.2.4. Motor Activity Log–Amount of Use Scale (MAL-AoU)

The MAL-AoU is a questionnaire that measures perceived UL activity performance in daily life for persons with neurologic conditions. The MAL-AoU was used here as a condition-specific tool for quantifying UL activity performance in daily life by participants with stroke and multiple sclerosis. The MAL-AoU asks participants to report the amount of UL use in 30 functional activities. Each item is scored on a 6-point ordinal scale that ranges from 0 (“I did not use my affected arm”) to 5 (“I used my affected arm as much as before my condition [stroke, multiple sclerosis]).” Item scores are averaged, with overall scores ranging from 0 to 5, with higher scores indicating greater reported use of the affected UL in daily life. The MAL has excellent psychometric properties for test–retest reliability and internal consistency as well as criterion validity [31,32].

#### 2.2.5. Disability of the Arm, Shoulder, and Hand Scale (DASH)

The DASH scale is a 30-item questionnaire that measures perceived UL activity performance in daily life for persons with musculoskeletal conditions. The DASH scale was used here as a condition-specific tool for quantifying UL activity performance in daily life from the participants with UL fractures, shoulder pain, and breast cancer. The DASH survey has 30 items, and each item is rated on a 5-point Likert scale where 1 means no difficulty with the task and 5 means being unable to perform that task. Items scores are summed with total scores range from 0 to 100, with lower scores indicating less disability. The DASH has excellent test–retest reliability, internal consistency, and construct validity [33,34,35].

#### 2.2.6. Activity Card Sort Test (ACS)

The ACS measures participation in four domains of activities across daily life: instrumental, social, low-demand physical leisure and high-demand physical leisure [36,37]. The ACS was used here to quantify participant perception of return to activities involving the UL. Participants are shown pictures of various activities and asked to rate each activity as either: 0 meaning they never engaged in the activity, or they gave up on the activity after their diagnosis; 0.5 meaning that they partially engage in that activity now since their diagnosis; or 1 meaning that they continue to engage in that activity or have now started to engage in that activity since their diagnosis. For this study, we used 65 of 89 activities that involve the UL, as done previously in a clinical trial [38,39]. We were most interested in the ACS Global Score encompassing all four domains, as well as the ACS Instrumental Activities of Daily Living (ACS IADL) score, which here was a subset of 17 items. The ACS has been used across a wide variety of chronic diseases and conditions and in the general population, and shows acceptable to excellent test–retest reliability and internal consistency [40,41,42,43].

#### 2.2.7. European Quality of Life Scale—5 Dimensions 3 Levels (EuroQoL)

The EuroQoL is a descriptive, standardized self-report measure of overall quality of life in 5 dimensions: mobility, self-care, usual activities, pain/discomfort, and anxiety/depression. The EuroQoL self-care scale was used here to capture aspects of UL activity performance in daily life, with the rationale that most self-care activities require the use of the ULs. There are 5 items on this scale; participants rate each item as presenting no problems, some problems, or more extreme problems. Lower scores indicate better function. The EuroQoL has been used across a wide variety of chronic diseases and conditions and in the general population [44]. The EuroQoL has excellent psychometric properties for test–retest reliability and convergent validity [44].

#### 2.2.8. Demographic and Other Data

Demographic and descriptive data were obtained by self-report or, for participants in clinical subgroups, by review of electronic medical records or from the referring medical providers when available. Additional self-report measures were collected to describe comorbid status (Charlson Comorbidity Index [45,46], score range = 0–29, higher scores = more disease burden) and depressive symptomatology (Center for Epidemiological Studies Depression Scale (CES-D), score range = 0–60, higher scores = greater depressive symptoms [47,48,49,50]). For this study, the CES-D was also used as a tool to confirm divergent/discriminant validity of the accelerometry measures.

### 2.3. Statistical Analysis

#### 2.3.1. Software Used and Data Availability Statement

All data were managed and analyzed in R (version 4.4.2), an open-source statistical computing software [20]. Data were managed with the *tidyverse* [51] package, analyzed using the *factoextra* [52] and *stats* [20] packages, and then visualized using both the *tidyverse* [51] and *patchwork* [53] packages in R. De-identified data are available from the lead authors upon request. Once the overall study is completed, all data will be publicly shared through the NIH-NICHD Data and Specimen Hub (DASH) repository. R code for all statistical analyses is available from the lead author’s GitHub repository: https://github.com/cem2183/sensor_categories (accessed on 11 June 2025).

#### 2.3.2. Sample Size

Determination of sample size for this study was based on a sensitivity analysis informed by preliminary data in a sample of people with stroke and people without UL disability that yielded a 5-variable, 5-cluster solution [11]. Simulations showed that *N* ≥ 200 would yield >80% classification accuracy to detect a range of differences between 5 cluster centroids across 5 standardized variables [54].

#### 2.3.3. Replication and Generalizability Analyses

The first two purposes of this study were to (1) replicate prior findings in a new sample of people with stroke and people without UL disability and (2) generalize those findings beyond stroke to other conditions for which people seek out UL rehabilitation. Multiple principal component analyses (PCAs) and cluster analyses were used to achieve these purposes. As such, the replication analysis was performed first with a sample of people with stroke (*n* = 49) and people without UL disability who served as controls (*n* = 138). A second replication analysis was performed on a sample that had matching stroke (64%) and proportionate control (36%) to the sample from Barth et al. (2021) [11]. The second sample retained all the participants with stroke (*n* = 49), while people without UL disability were sampled to produce an age-matched distribution (*n* = 20). After replication was confirmed, analytic methods were repeated using a third sample of people with distal UL fracture, proximal UL pain, breast cancer, multiple sclerosis, and people without UL disability to determine initial generalizability of findings without people with stroke. Finally, the analysis was repeated on a fourth sample using all participants (*N* = 324).

#### 2.3.4. Principal Component Analyses

Principal components were derived from sets of accelerometry input variables (12, 9, 7, and 5). Prior to conducting PCAs, all accelerometry variables were standardized using z-scores due to different measurement scales (e.g., counts, hours, ratios). All 12 accelerometry input variables were then visualized in density plots stratified by diagnosis (Figure 1). To replicate work by Barth et al., 2021 [11], the PCA was conducted using the *prcomp* function as part of the *stats* [20] package on sets of 12, 9, 7, and then 5 accelerometry input variables, as indicated in the last column in Table 1. Accelerometry input variables were sequentially excluded from analyses according to computational complexity, validation in clinical populations, and clinical interpretation [11]. Additionally, each model had at least one accelerometry input variable from constructs of duration, intensity, variability, and symmetry to best capture the dimensionality of UL activity performance in daily life [11]. For each variable model (12, 9, 7, and 5), scree plots were evaluated to determine the appropriate number of PCs to explain variance among the accelerometry input variables. Importantly, because the sign of a PC is arbitrary, we manually set the direction of the PC to load positively on preferred time across all analyses (e.g., if the loading of PC1 on preferred time was negative, all loadings for PC1 were multiplied by −1). This alignment does not alter the internal structure of the loadings but ensures directional consistency across variable sets.

#### 2.3.5. Cluster Analyses

Following the PCAs, we used the *factoextra* [52] package to perform a k-means cluster analysis as an efficient means to identify potential subgroups of participants within the data (k tested from 1 to 10). Previously, a 5-variable, 5-cluster solution was sufficient for people with stroke and people without UL disability [11]. However, with replication and generalization to other conditions, we estimated that upwards of 8 clusters may be necessary to explain the data. We used several statistical methods to determine the most appropriate number of clusters for a given set of input variables: (1) the elbow method on a scree plot of the within cluster sum of squares (WSS) [55], (2) the silhouette statistic [56], and the (3) gap statistic [57]. Collectively, from the scree plot, silhouette statistic, and the gap statistic, 2–5 cluster solutions progressively explained the data, and so we focused on these cluster sizes for further analyses.

To adjudicate between 2- to 5-cluster solutions, we extracted cluster membership from each solution for each variable set (12, 9, 7, and 5 input variables). Treating cluster membership as a categorical factor, we assessed model fit via multivariate analysis of variance (MANOVA) via the *stats* [20] package. MANOVA produced the percentage of total variance explained by the number of clusters, allowing for the assessment of model fit [58]. To penalize for overfitting, MANOVA also allowed us to calculate the Akaike Information Criterion (AIC) via the *stats* package for each number of clusters and input variables [58]. The AIC imposes a penalty for additional model parameters, so the model with the lowest AIC value was chosen to avoid overfitting and enhance generalizability [58].

#### 2.3.6. Determining Convergent and Divergent Validity

The third purpose of this study was to evaluate the convergent validity between sensor-based categories, self-reported daily UL activity, and other related measures. The convergent validity of generic (i.e., PROMIS UE score) and condition-specific (i.e., MAL-AoU, DASH) self-report of UL activity performance and disability were compared against (1) 2 PCs, (2) the 5-cluster model, and then (3) individual accelerometry variables using the *lm* function of the *stats* [20] package to obtain R^2^. As some models rely on multiple predictors (e.g., multiple clusters or principal components) and other models use a single predictor (e.g., paretic arm time or the use ratio), we focused on the R^2^ of all models to understand how these self-report outcomes related to accelerometry-derived values. For convergent validity of single predictors, we regressed condition-specific outcomes (i.e., MAL-AoU, DASH) onto individual accelerometry measures. As a test of divergent validity, we regressed a measure of depressive symptoms (CES-D) onto individual accelerometry measures. Measures of depressive symptoms should have little to no relation to accelerometry measures [59].

## 3. Results

A total sample of *N* = 324 participants were included in the various analyses. Demographic and participant characteristics for the five clinical sub-groups and people without UL disability are provided in Table 2. Baseline scores across self-report measures are provided in Table 3. People with proximal UL pain generally had the highest incidence of concordance (where the dominant limb is the affected upper limb) followed by people with multiple sclerosis, distal UL fracture, stroke, and then breast cancer. Figure 1 displays density plots for each accelerometry input variable, by diagnosis.

Despite differing numbers of accelerometry input variables, two principal components (PC1, PC2) consistently explained the majority of variance for each sample in our analysis. Statistics from the PCs and two-, three-, four-, or five-cluster evaluations across 12, 9, 7, or 5 accelerometry input variables are displayed along with the variance explained and model fit criteria in Table 4. The most parsimonious model used five accelerometry input variables. Using this five-variable model, PC1 explained most of the variance for all samples (range 53.0–76.4) and consistently showed moderate loadings across all five accelerometry input variables. PC loadings represent the contribution of each original accelerometry input variable to the principal component. PC1 appears to capture overall UL activity level. PC2 explained considerably less variance across all samples (range 17.6–25.7). PC2 appears to reflect the duration of unilateral activity of the preferred limb (dominant limb in control participants, non-affected limb in patient participants), based on the strong (positive) loading response from preferred time (Figure 2).

A five-cluster solution consistently explained the most overall variance in comparison to two-, three-, or four-cluster solutions for each sample in our analysis, regardless of the number of accelerometry input variables. Cluster statistics are shown in the last two columns of Table 4 and represented graphically in Figure 3. For samples that included participants with stroke and people without UL disability, the analyses replicated prior work [11]. AIC values (last column, Table 4) determined that a five-variable five-cluster solution provided the best model fit in comparison to the two-, three-, or four- cluster solutions across different accelerometry input variables (12, 9, 7 and 5). This held consistent across all samples. While each cluster solution (2-, 3-, 4-, 5-) was statistically feasible, the final, most parsimonious solution included five clusters, with five accelerometry input variables of non-preferred magnitude, non-preferred time, non-preferred variance, preferred time, and the use ratio, consistent with prior work. Figure 3 displays a five-variable, five-cluster solution within a two-dimensional PC space for each of the four samples in this analysis. The *x*-axis represents PC1, which captures overall UL activity, while the *y*-axis represents PC2, which represents the duration of preferred limb use. Each axis reflects a continuous gradient, with participants distributed according to their clustered UL performance profiles. To aid interpretation, consider the following examples in Figure 3: Cluster 1 (shown in blue) comprises individuals with high preferred UL use but low overall UL activity, suggesting limited engagement of the non-preferred limb in daily tasks, and likely greater severity of UL functional deficits. Cluster 5 (shown in purple includes individuals with both high preferred UL use and high overall UL activity, consistent with more symmetrical and frequent UL use throughout the day, and likely less severe or no impairments. Cluster 2 (shown in orange) consists of individuals with low preferred UL use and low-to-moderate overall UL activity, indicating either impairment with the preferred UL, or reduced overall upper limb engagement. Overall, clusters show remarkable similarities across each sample, despite differing compositions of clinical populations and severity of functional deficits.

Surprisingly, there was better convergent validity of the two continuous PCs and individual continuous UL accelerometry variables compared to the five categorical subgroups from our clustering algorithm. Continuous variables generally explained more variance in generic and condition-specific self-report outcomes of UL activity performance and disability, compared to the categorical clusters. Figure 4 visually illustrates this by showing the lack of relationship between the cluster solutions and the PROMIS Upper Extremity measure and the ACS IADL measure in the top row (Figure 4A,B) compared to scatter plots of PC 1 (Figure 4C,D) and example single variables (Figure 4E,H). Table 5 shows the R^2^ values that quantify these relationships. Of note, there are no universally defined thresholds for interpreting R^2^ values; rather, they should be evaluated in relation to the construct being measured and the methods used. In the context of this work, each accelerometry input variable captures a single aspect of a broader construct being that of performance of UL activity in daily life. Accordingly, modest R^2^ values across multiple variables may still provide meaningful evidence of convergent validity when considered collectively. For example, many single UL accelerometry variables (e.g., non-preferred magnitude, bilateral magnitude, non-preferred variance, use ratio, and magnitude ratio) showed convergent validity with self-report outcomes of UL activity performance (column 2, Table 5) and UL disability (columns 5–6, Table 5). Three UL accelerometry variables (e.g., spectral arc length and preferred/non-preferred-only times) consistently showed the weakest convergent validity across self-report outcomes of UL activity performance and disability. Evaluation of condition-specific measures revealed no convergent validity across UL accelerometry variables for the DASH scale. However, we observed significant relations for convergent validity on the MAL (e.g., non-preferred magnitude, bilateral magnitude, non-preferred variance, use ratio, magnitude ratio, and jerk asymmetry index). Additionally, the relationship between the PROMIS Upper Extremity measure and the two PCs is represented as an R^2^ of 0.190 indicating significance, while clusters showed essentially no relationship with this measure as an R^2^ 0.030. Divergent validity was established using the CES-D, wherein low values of R^2^ were observed across five clusters, two PCs, and individual UL accelerometry variables.

## 4. Discussion

A five-variable, five-cluster solution for UL activity performance in daily life was replicated in a new independent sample of people with stroke and people without UL disability. Furthermore, the relationships between variables (PCA) and clustering of people into roughly five multivariate groups (k-means), generalized across neurological, musculoskeletal, and other medical conditions and across severity of functional deficits, were studied. Expanding the analyses from stroke to other conditions resulted in remarkable similarities across samples. This suggests that UL activity performance in daily life can be quantified in a similar manner regardless of the biological cause or severity of functional deficits. However, across generic and condition-specific self-report outcomes of UL activity performance and disability, convergent validity was much higher for PCs and individual UL accelerometry variables treated continuously than for the five clusters treated categorically. Thus, although there is empirical support to divide people into clusters, when it comes to explaining individual differences in UL activity performance and disability, treating these measures continuously may be the more powerful approach. Overall, these findings underscore the potential for more efficient and personalized rehabilitative care by streamlining wearable sensor-based assessment of UL activity performance into a focused, generalizable tool that is applicable across diverse conditions of the UL as well as severity of functional deficits and salient to clinicians and patients alike.

Replication of a five-variable, five-cluster solution in a sample of people with stroke and people without UL disability was confirmed. This analysis demonstrated the stability of original findings from Barth et al. (2021) [11] across two new samples of people with stroke and people without UL disability. Although both replication samples included the same conditions, their compositions differed substantially, which provided an opportunity to assess the robustness of the model across contrasting clinical profiles. Irrespective of sample composition (one sample had a higher frequency of people without UL disability than the other), two PCs consistently emerged from the data with comparable variance explained, and the five-variable, five-cluster model was preserved (samples 1 and 2, Table 4). Replication of the same solution suggests the findings are real, as many findings in the biomedical literature cannot be replicated [60,61], especially with statistical models such as the ones used here.

A five-variable, five-cluster solution was generalizable beyond people with stroke and people without UL disability, to people with other neurologic, musculoskeletal, and medical conditions that affect the ULs to varying degrees of functional severity. This generalizability is visually supported by the consistent spatial patterns observed across samples in both the PC space (Figure 2) and cluster distributions (Figure 3). There are two intertwined reasons that may account for this generalizability. First, all people need to perform similarly complex UL activities in daily life. And second, accelerometry-derived sensor variables quantify general characteristics of movement (e.g., duration, magnitude, quality) [62], but they do not yet quantify how and when specific tasks are performed by a person [18]. While efforts are underway in other research groups to quantify specific, individual tasks performed in daily life, it may be decades before the library of algorithms is large and accurate enough to be used in an unsupervised clinical setting across patient populations and severity of functional deficits [63,64,65,66]. Taken together, the generalizability observed for both PCA and cluster analyses suggests a practical and scalable solution for characterizing UL activity performance in daily life across diverse clinical conditions and severity of functional deficits.

Assessment of convergent validity revealed PC and individual UL accelerometry variables had significant relationships with both generic and condition-specific self-report measures of UL activity performance and disability, while cluster membership did not. Five categorical clusters showed trivial relationships with PROMIS UE and ACS IADL scores (Figure 4A,B; R^2^ ≤ 0.03), suggesting that although these people may consistently move similarly based on their accelerometry recorded in daily life, cluster membership was unrelated to perceptions of UL activity in daily life. In contrast, PCs showed significant relations with self-report measures (Figure 4C,D), particularly condition-specific instruments like the MAL (R^2^ = 0.26–0.27), and broader indices of activity and participation (R^2^ = 0.26–0.34). While the data support consistent groups of people with similar movement profiles, there remains important variability within each group. This variability is meaningfully associated with clinically validated outcomes but is lost when continuous data are reduced to categorical classifications. Similarly, among individual accelerometry variables, those reflecting movement magnitude (e.g., bilateral magnitude) and symmetry (e.g., magnitude ratio, jerk asymmetry index) showed significant relations with self-reported activity (R^2^ up to 0.27), while duration (e.g., preferred/non-preferred only time) and quality of movement (e.g., spectral arc length) did not. The magnitude of the significant relationships aligns with prior findings in people with stroke, though it is presented here as R^2^ rather than correlation coefficients [66], to allow comparison with multivariable constructs such as the categorical clusters and two PCs. Overall, these findings suggest that upper limb performance may be better represented along a continuum of functional recovery using 2PCs, rather than with discrete categories using clusters. In the future, a five-variable model could be implemented clinically using onboard processing in either wearable movement sensors or mobile devices to map an individual’s performance within a two-dimensional PC space. This approach may offer several benefits: it can be generalized across conditions and severity of functional deficits, avoids reliance on condition-specific clinical assessments, and reduces the need for recall of diagnostic-specific details. However, certain clinical populations may still benefit from targeted examination of specific variables, for example, the use ratio in stroke or jerk asymmetry in ataxia, highlighting the potential need for hybrid approaches in some contexts or environments.

This study had two primary limitations, which influence interpretation of the data. First, the proportion of participants people without UL disability relative to individual clinical groups was high. While the total sample was predominantly composed of clinical populations (57%), the relatively high proportion of people without UL disability (43%) may have biased results towards lower levels of disability. The inclusion of people without UL disability, however, served a critical role by providing a reference point for normative functional status, given that the aims of this work were to replicate and generalize prior findings to additional clinical populations. Second, the aim of replication required use of the same composition of (12, 9, 7, and 5) accelerometry input variables for each stage of analysis. This approach may have limited discovery of other relevant variables or combinations that could reveal additional patterns across clinical populations and severity of functional deficits. This limitation was further amplified in the assessment of convergent validity, where some individual accelerometry variables demonstrated comparable associations with self-reported outcomes as the multivariate PCs. It is possible that other, multivariate combinations could be identified in the future and might be more strongly related to perception of UL activity performance in daily life.

## 5. Conclusions

This work demonstrates that while a five-variable, five-cluster solution can be reliably reproduced and generalized across a diverse group of clinical populations and severity of functional deficits, it does not fully capture the clinically meaningful variation of UL activity performance in daily life. Stronger associations between self-reported outcomes and both PC and individual accelerometry variables indicated that key aspects of functional recovery are lost when continuous data are reduced to categories. These findings support a shift toward models that quantify UL recovery along a continuum, which may offer greater sensitivity to individual differences and broader applicability across diagnostic groups and functional levels of severity. This approach lays the foundation for the development of efficient and scalable tools to monitor UL function in real-world rehabilitation settings.

## Figures and Tables

**Figure 1 sensors-25-04618-f001:**
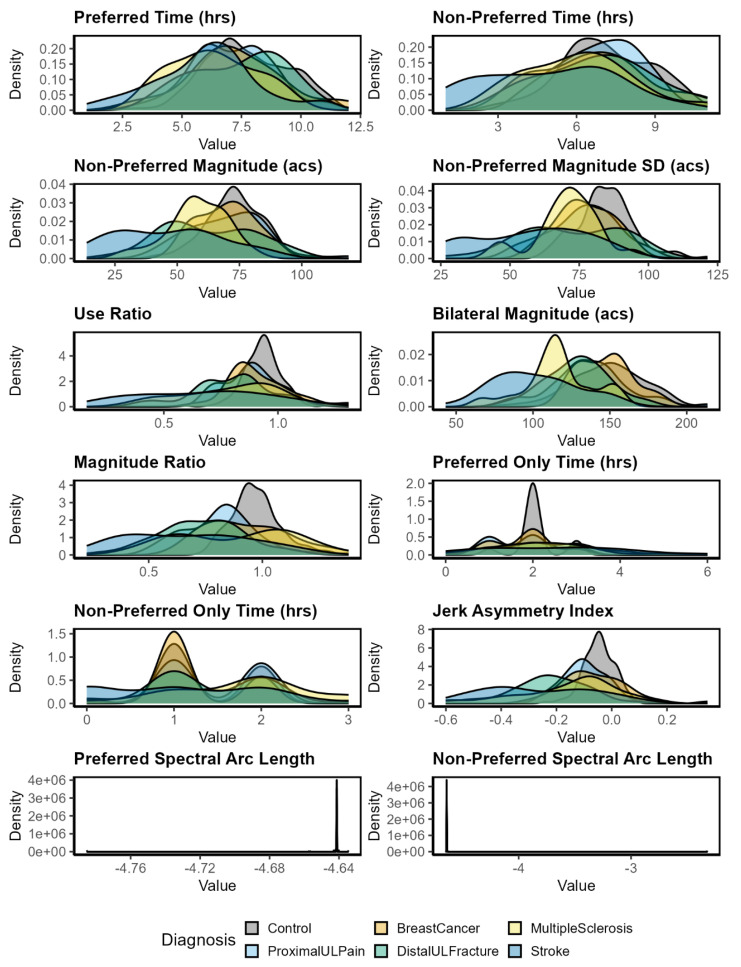
Density plots for each accelerometry input variable by diagnosis. Abbreviations: acs, activity counts; hrs, hours; ProximalULPain, proximal upper limb pain; DistalULFracture, distal upper limb fracture.

**Figure 2 sensors-25-04618-f002:**
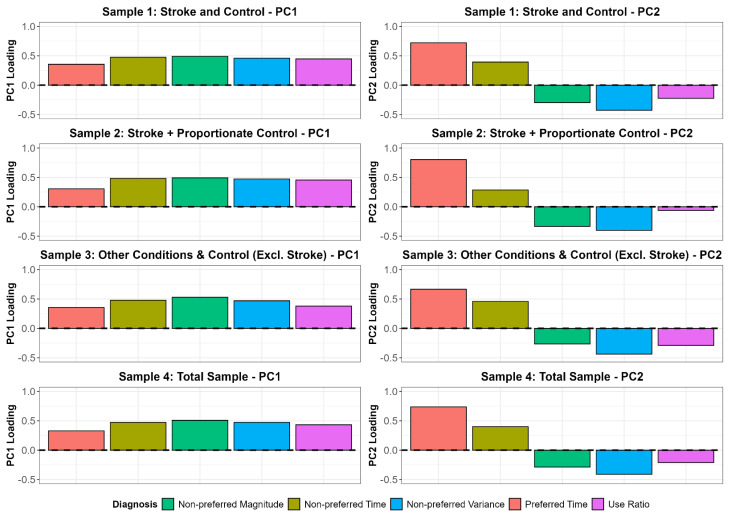
Principal component variance explained. Abbreviations: Excl, excluding; PC, principal component.

**Figure 3 sensors-25-04618-f003:**
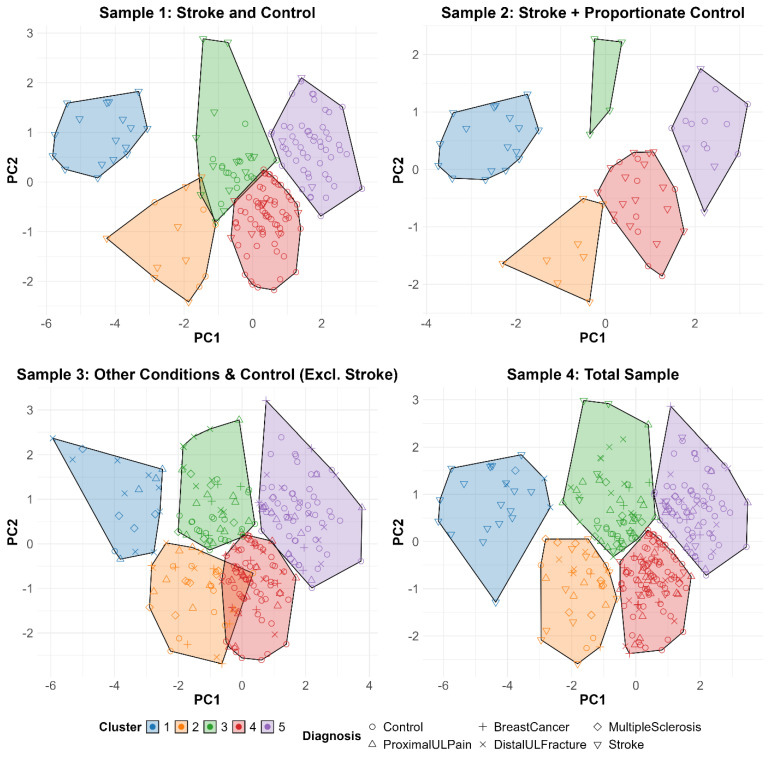
Five-variable, five-cluster plot across samples. Abbreviations: Excl., excluding; PC, principal component; ProximalULPain, proximal upper limb pain; DistalULFracture, distal upper limb fracture.

**Figure 4 sensors-25-04618-f004:**
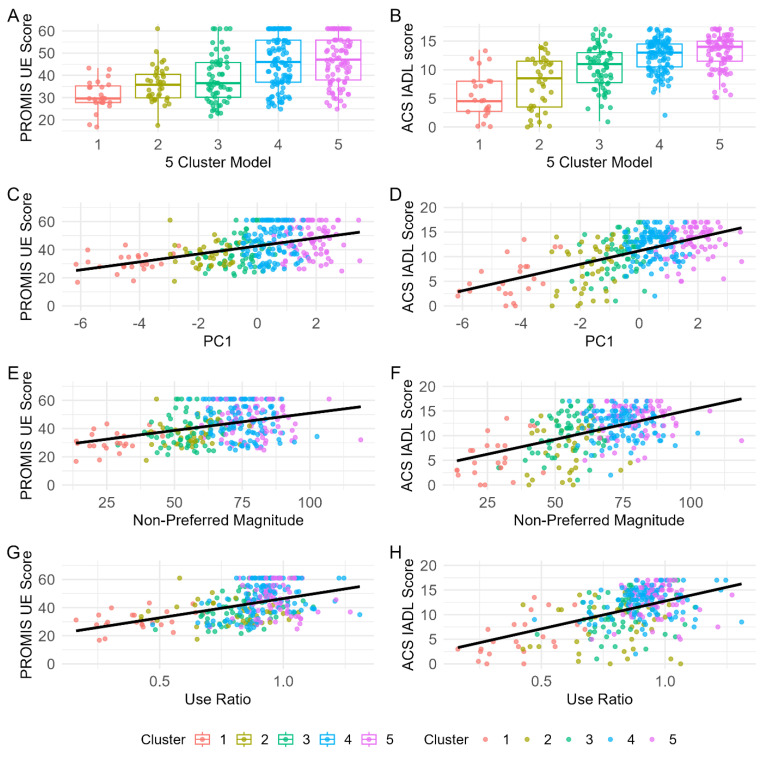
Scatterplots of convergent validity analysis. (**A**) Scatterplot of PROMIS UE score and 5 Cluster Model. (**B**) Scatterplot of ACS IADL score and 5 Cluster Model. (**C**) Scatterplot of PROMIS UE and PC1 (which represents overall UL activity performance). (**D**) Scatterplot of ACS IADL score and PC1 (which represents overall UL activity performance). (**E**) Scatterplot of PROMIS UE score and Non-Preferred Magnitude. (**F**) Scatterplot of ACS IADL score and Non-Preferred Magnitude. (**G**) Scatterplot of PROMIS UE score and Use Ratio. (**H**) Scatterplot of ACS IADL score and Use Ratio. Abbreviations: ACS IADL score, Activity Card Sort–Instrumental Activities of Daily Living score; PC, principal component; PROMIS UE Score, Patient-Reported Outcome Measurement Information System Score for the Upper Extremity.

**Table 1 sensors-25-04618-t001:** Description of UL wearable movement sensor variables, adapted from Barth et al. (2021) [11].

	Data Source	UL Accelerometry Performance Variable Name	Description and Interpretation	Accelerometry Input Variable Set
Duration (h)	1 Hz	Preferred time	Total time that the preferred limb is moving, as determined by activity counts > 2 for each second [24].	12, 9, 7, 5
	1 Hz	Non-preferred time	Total time that the non-preferred limb is moving, as above.	12, 9, 7, 5
	1 Hz	Preferred-only time	Total time that only the preferred limb is moving, as above.	12, 9
	1 Hz	Non-preferred-only time	Total time that only the non-preferred limb is moving, as above.	12, 9
Intensity (acs)	1 Hz	Non-preferred magnitude	Median of the accelerations for the non-preferred limb when it was moving (excluding non-movement time). Higher movement counts indicate greater movement intensity of the non-preferred limb.	12, 9, 7
	1 Hz	Bilateral magnitude	Sum of the non-preferred and preferred magnitudes, as above. Higher activity counts indicate greater intensity of movement across both limbs.	12, 9, 7
Variability	1 Hz	Non-preferred variance	Standard deviation of the magnitude of accelerations for the non-preferred limb when it was moving. Higher activity counts indicate greater movement variability of movement for the non-preferred limb.	12, 9, 7, 5
	1 Hz	Use ratio	Ratio of hours of non-preferred time to preferred time. Values are generally between 0 and 1, with values close to 1 indicating equal integration of both limbs into daily activities.	12, 9, 7, 5
	1 Hz	Magnitude ratio	Ratio of the magnitude of non-preferred versus preferred limb accelerations (intensity). Interpretation as above, except with magnitudes instead of durations.	12, 9, 7
Symmetry	30 Hz	Jerk asymmetry index [25]	Ratio of the jerk of the non-preferred and preferred limbs but calculated as ((jerk_non-preferred_ − jerk_preferred_)/jerk_non-preferred_ + jerk_preferred_)). Values range from −1 and +1, with values around 0 indicating similar smoothness of movement in the limbs. Values closer to +1 or −1 reflect greater jerk for the non-preferred limb and the preferred limb, respectively.	12
Movement Quality	30 Hz	Preferred spectral arc length [26,27]	Measurement of the “arc length” of the Fourier magnitude spectrum within a certain frequency range. This measure is independent of the movement’s amplitude and duration and indicates smoothness of movement by quantifying movement interruptions. More negative spectral arc lengths are reflective of less smooth or less coordinated movement in the preferred/non-preferred limbs.	12
	30 Hz	Non-preferred spectral arc length		12

Abbreviations: acs, activity counts; h, hours. Legend: The *Accelerometry Input Variable Set* column describes groups of single accelerometry input variables that were systematically reduced based on perceived clinical utility. These variable sets are based on prior work by Barth et al., 2021 [11].

**Table 2 sensors-25-04618-t002:** Demographics of the sample. Values are presented as percentage [*n*] or median [IQR].

		Total Sample (n = 324)	People Without UL Disability (n = 138)	People with Stroke (n = 49)	People with Proximal UL Pain (n= 55)	People with Distal UL Fracture (n= 40)	People with Breast Cancer (n= 23)	People with Multiple Sclerosis (n= 19)
Age		53 [40, 67]	41 [29, 60]	59 [52, 70]	59 [49, 68]	63 [51, 68]	56 [43, 64]	49 [43, 53]
Sex	Male	28% [91]	25% [34]	59% [29]	35% [19]	12% [5]	NA	21% [4]
	Female	72% [233]	75% [104]	41% [20]	65% [36]	88% [35]	100% [23]	79% [15]
Race	American Indian or Alaska Native	<1% [1]	<1% [1]	NA	NA	NA	NA	NA
	Asian	6% [21]	12% [17]	NA	5% [3]	NA	NA	5% [1]
	Black or African American	23% [76]	20% [28]	43% [21]	20% [11]	3% [1]	17% [4]	58% [11]
	Native Hawaiian or Other Pacific Islander	<1% [3]	NA	NA	NA	3% [1]	NA	5% [1]
	White	69% [224]	67% [92]	57% [28]	75% [41]	94% [38]	83% [19]	32% [6]
Ethnicity	Hispanic, Latinx	5% [15]	4% [6]	8% [4]	8% [4]	6% [2]	9% [2]	6% [1]
	Non-Hispanic, Non-Latinx	99% [309]	96% [132]	92% [45]	82% [45]	94% [38]	91% [21]	94% [16]
Employment Status	Not working for paid employment	46% [148]	31% [43]	86% [42]	36% [20]	42% [17]	48% [11]	79% [15]
	Working < 20 h/week	8% [25]	11% [15]	2% [1]	7% [4]	10% [4]	4% [1]	NA
	Working part-time ≥ 20 h/week	6% [18]	6% [8]	4% [2]	4% [2]	10% [4]	4% [1]	5% [1]
	Woking full-time ≥ 37.5 h/week	40% [133]	52% [72]	8% [4]	53% [29]	38% [15]	44% [10]	16% [3]
Hand Dominance	Right	90% [292]	93% [129]	88% [43]	93% [51]	85% [34]	78% [18]	89% [17]
	Left	9% [29]	7% [9]	10% [5]	7% [4]	10% [4]	22% [5]	11% [2]
	Ambidextrous	1% [3]	NA	2% [1]	NA	5% [2]	NA	NA
Affected Side	Right	NA	NA	45% [22]	56% [31]	42% [17]	52% [12]	53% [10]
	Left	NA	NA	55% [27]	44% [24]	58% [23]	47% [11]	47% [9]
Time Since UL Dysfunction/Pain		NA	NA	3 mo [1.5, 12]	2 yrs [1, 4]	1.6 mo [1.1, 1.8]	12 mo [5.75, 30]	13 yrs [8, 22]
Concordance *	Yes	NA	NA	41% [20]	53% [29]	42% [17]	39% [9]	42% [8]
	No	NA	NA	59% [29]	47% [26]	58% [23]	61% [14]	58% [11]
Total Charleson Comorbidity Index Score		1 [0, 3]	1 [0, 3]	3 [2, 4]	3 [1, 4]	3 [1, 3]	3 [2, 4]	2 [1, 2]
Average Accelerometry Weartime †		100% [324]	100% [138]	100% [49]	100% [55]	100% [40]	100% [23]	100% [19]

* Concordance: dominant limb = paretic limb. † Adherence to wearing for this cohort was 96% for at least a single day. Data from participants with <1 day of recording were excluded from this report. Abbreviations: ADL, Activities of Daily Living; IQR, inter-quartile range; mo, months; *n*, number of participants; NA, Not Applicable; SD, standard deviation; yrs, years.

**Table 3 sensors-25-04618-t003:** Baseline data for self-report measures. Values are median [IQR].

Self-Report Measure (Points)	Total Sample	People Without UL Disability	People with Stroke	People with Proximal UL Pain	People with Distal UL Fracture	People with Breast Cancer	People with Multiple Sclerosis
PROMIS Upper Extremity Score	42 [32, 52]	55 [47, 61]	34 [29, 39]	36 [30, 38]	32 [28, 38]	37 [34, 44]	32 [27, 39]
MAL—Amount of Use Scale Score	NA	NA	3 [2, 4]	NA	NA	NA	2 [2, 4]
DASH Score	NA	NA	NA	40 [23, 53]	41 [30, 60]	35 [9, 49]	NA
ACS Global Score	31 [24, 38]	36 [31, 43]	21 [15, 26]	30 [25, 34]	27 [22, 34]	31 [23, 40]	21 [15, 28]
ACS IADL Score	12 [9, 14]	14 [12, 15]	6 [3, 9]	13 [9, 15]	11 [8, 13]	12 [10, 14]	7 [3, 9]
Euro-QofL—Self Care Score	1 [1, 1]	1 [1, 1]	2 [1, 2]	1 [1, 2]	1 [1, 2]	1 [1, 1]	1 [1, 2]
Euro-QofL—Usual Activities Score	1 [1, 2]	1 [1, 1]	2 [2, 2]	2 [1, 2]	2 [2, 2]	2 [1, 2]	2 [2, 2]
CES-D Score	9 [4, 17]	8 [4, 15]	13 [8, 23]	8 [3, 15]	7 [3, 13]	11 [5, 16]	12 [8, 26]

Abbreviations: ACS, Activity Card Sort; CES-D, Center for Epidemiological Studies–Depression Scale; DASH, Disabilities of the Arm, Shoulder, and Hand Scale; Euro-QofL, European Quality of Life Scale; IADL, Instrumental Activities of Daily Living; MAL, Motor Activity Log; NA, Not Applicable; PROMIS, Patient-Reported Outcome Measure Information System; IQR, inter-quartile range; *n*, number.

**Table 4 sensors-25-04618-t004:** Cluster statistics are shown per sample sub-set and listed in ascending order by accelerometry input variable. Values are to be compared within each row of data.

Sample	Number of Accelerometry Input Variables	Variance Explained by Each PC (%)		Total Variance Explained by Number of Clusters (%)				AIC by # of Clusters			
		PC1	PC2	2	3	4	5	2	3	4	5
Sample 1: Stroke + Control (n = 192, replication)	12	57.4	13.1	35.9	45.2	53.8	59.4	1462.7	1281.9	1116.6	1017.2
	9	68.5	16.5	40.6	53.2	59.8	64.6	1019.7	829.9	738.1	677.0
	7	75.6	14.1	46.8	61.4	67.7	70.8	713.3	539.4	471.6	445.7
	5	76.4	17.6	45.7	62.7	69.4	73.9	519.6	373.1	321.6	289.7
Sample 2: Stroke + Proportionate Control (n = 69, replication)	12	49.6	13.2	39.5	48.5	55.3	60.1	527.0	479.9	450.4	436.1
	9	58.9	17.5	46.6	58.1	65.2	69.7	353.4	302.7	278.6	270.3
	7	66.9	15.0	51.1	63.0	68.9	73.6	254.0	213.0	199.8	192.1
	5	67.1	19.5	49.9	64.8	70.9	75.8	185.3	146.1	136.0	129.9
Sample 3: Other conditions +Control excluding stroke (n = 275, generalization)	12	34.2	15.5	20.9	30.6	39.4	46.4	2638.5	2343.9	2080.2	1876.1
	9	40.4	19.8	25.5	37.4	45.1	50.5	1865.4	1593.4	1420.4	1306.5
	7	49.5	21.2	31.2	44.3	51.1	56.0	1343.4	1107.4	990.7	911.7
	5	53.0	25.7	58.0	72.1	78.2	81.6	935.4	720.6	645.5	588.0
Sample 4: Total Sample (n = 324, generalization)	12	42.9	13.7	28.0	37.9	46.5	52.2	2823.4	2465.7	2158.4	1868.0
	9	50.6	18.0	32.1	45.0	51.9	62.1	1997.2	1643.2	1460.7	1305.2
	7	59.5	17.5	37.2	51.4	58.7	63.7	1440.0	1134.8	984.2	885.9
	5	67.6	19.5	38.4	53.8	62.7	68.0	1008.1	772.1	637.7	563.5

Abbreviations: Akaike Information Criterion (AIC), principal component (PC).

**Table 5 sensors-25-04618-t005:** Convergent and divergent validity statistics.

Accelerometry Input Variable (R^2^)	Universal Self-Report of UL Activity	Condition Specific Self-Report of UL Activity		Common Self-Report of Activity and Quality of Life				Self-Report of Depressive Symptoms
	PROMIS UE	MAL (Stroke, MS)	DASH (Breast Cancer, Distal UL Fracture, Proximal UL Pain)	ACS Global	ACS IADL	Euro QofL Self-Care	Euro QofL Usual Activities	CES-D
5 Clusters	0.030	**0.210**	**0.150**	0.015	0.027	0.065	0.025	0.007
2 PCs	**0.190**	**0.260**	0.010	**0.260**	**0.340**	**0.150**	**0.140**	0.063
Preferred time	0.036	0.057	0.002	**0.144**	**0.152**	0.053	0.048	0.040
Non-preferred time	0.057	0.012	0.023	**0.152**	**0.116**	0.068	0.063	0.063
Preferred only time	4.0 × 10^−4^	0.053	0.048	0.012	0.026	0.003	2.5 × 10^−5^	4.0 × 10^−6^
Non-preferred only time	0.004	0.053	0.004	0.004	0.023	4.9 × 10^−5^	0.004	0.005
Non-preferred magnitude	**0.130**	**0.212**	4.0 × 10^−4^	**0.168**	**0.260**	0.090	0.090	0.005
Bilateral magnitude	**0.203**	**0.194**	0.068	**0.240**	**0.325**	**0.152**	**0.152**	0.014
Non-preferred variance	**0.194**	**0.203**	0.006	**0.176**	**0.270**	**0.109**	**0.110**	0.004
Use ratio	**0.176**	**0.260**	0.058	**0.176**	**0.260**	**0.160**	**0.144**	1.0 × 10^−4^
Magnitude ratio	**0.212**	**0.230**	0.090	**0.102**	**0.144**	**0.102**	**0.116**	2.0 × 10^−4^
Jerk asymmetry index	**0.212**	**0.270**	0.090	**0.144**	**0.203**	**0.144**	**0.168**	1.0 × 10^−4^
Preferred spectral arc length	1.0 × 10^−4^	4.0 × 10^−4^	0.008	0.030	0.003	1.6 × 10^−4^	1.6 × 10^−4^	1.6 × 10^−4^
Non-preferred spectral arc length	2.0 × 10^−4^	0.040	0.017	9.0 × 10^−4^	4.0 × 10^−4^	1.6 × 10^−4^	1.6 × 10^−5^	4.9 × 10^−4^

Values in bold are statistically significant from zero. Preferred limb refers to the dominant limb in participants without UL disability, and the non-affected limb in participants with clinical conditions. Non-preferred limb refers to the non-dominant limb in participants without UL disability and the affected limb in participants with clinical conditions. Abbreviations: ACS, Activity Card Sort; CES-D, Center for Epidemiologic Studies Depression Scale; DASH, Disability of the Arm, Shoulder, and Hand Scale; EuroQoL, European Quality of Life Scale; IADL, Instrumental Activities of Daily Living; MAL, Motor Activity Log; Multiple Sclerosis; PROMIS, Patient-Reported Outcomes Measurement Information System; UL, upper limb.

## Data Availability

Informed consent was obtained from all subjects involved in the study.

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
