# Peer review of "Replication of Sensor-Based Categorization of Upper-Limb Performance in Daily Life in People Post Stroke and Generalizability to Other Populations"

_sensors, 2025, doi:10.3390/s25154618_

Round 1
Reviewer 1 Report
Comments and Suggestions for Authors
This study replicated and generalized a five-variable, five-cluster model of upper-limb (UL) activity performance in daily life across diverse clinical populations (stroke, MS, fractures, pain, breast cancer) and controls. Using accelerometry and principal component analysis, the model demonstrated robust applicability across conditions. However, convergent validity analyses revealed that continuous principal components and individual accelerometry variables showed stronger correlations with self-reported UL function and disability than categorical clusters. The findings support a continuum-based approach for monitoring UL recovery in rehabilitation, offering greater sensitivity and clinical utility than discrete categorization. This paper is well-structured and well-written. Before publication, there are some questions to be solved.
- The authors mentioned “In recent years, wearable sensors have emerged as a research tool to quantify UL activity performance in daily life and their potential for integration in clinical rehabilitation continues to grow.”, more state-of-the-art can be cited: DOI: 10.34133/cbsystems.0094; DOI: 10.34133/cbsystems.0243.
- What’s the meaning of the functional meaning of each of the 5 clusters (e.g., "high bilateral use with low intensity"), how these differ from the principal components?
- How about the robustness across different subsamples (stroke-only, mixed clinical, controls)?
- For the convergent validity thresholds, what R² thresholds are considered meaningful?
- The sample has few participants with severe UL impairment, and how might the model perform in this subgroup.
- How to implement the 5-variable model in clinical settings, are there any practical barriers?
- Did the authors test divergent validity with other unrelated constructs to further validate the specificity of accelerometry measures?
Author Response
All changes to the text are in red, and line numbers correspond to the revised manuscript.
1. The authors mentioned “In recent years, wearable sensors have emerged as a research tool to quantify UL activity performance in daily life and their potential for integration in clinical rehabilitation continues to grow.”, more state-of-the-art can be cited: DOI: 10.34133/cbsystems.0094; DOI: 10.34133/cbsystems.0243.
Thank you for these suggestions. To maintain the focus of this article on wearable motion sensing such as accelerometers, we have opted not to add the suggested references. Instead, we have revised our wording to clarify that our manuscript focus is specifically on accelerometry (line 72).
2. What’s the meaning of the functional meaning of each of the 5 clusters (e.g., "high bilateral use with low intensity"), how these differ from the principal components?
Thank you for this comment. The principal components were derived from sets of accelerometry variables (12, 9, 7, and 5) to capture multivariate dimensions of upper limb performance of activity in daily life. As we found, PC1 likely represented overall upper limb activity level, while PC2 likely represented the duration of the preferred limb (dominant limb in control populations, non-affected limb in patient populations). PC loadings represent the contribution of each original accelerometry input variable to the principal component (Figure 2).
In contrast, cluster analysis was used to identify subgroups within each sample (Stroke and control, Stroke and proportionate control, other populations excluding stroke and control, and total sample). While the clusters are categorical, they align along the PC1 and PC2 space, capturing distinct patterns of performance of upper limb activity in daily life (shown in Figure 3). For example, Cluster 1 (presented in blue, Figure 3) includes individuals with high preferred limb use (as evidenced by the positive location in PC2 space) but low bilateral activity (as evidenced by the negative location in PC1 space), suggesting limited integration of the non-preferred limb in daily tasks. Alternatively, Cluster 5, (presented in purple, Figure 3) shows individuals who have both high preferred arm use (as evidence by the positive location in PC2 space) and high overall upper limb activity (as evidenced by the positive location in PC1 space) suggesting that these individuals may have more symmetric and frequent upper limb use. Cluster 2 (presented in orange, Figure 3) includes individuals with low preferred limb use and low-to-moderate overall activity, potentially reflecting impairment in the preferred limb and/or generally less activity for the upper limbs. Our cluster numbers and arrangements across samples replicate prior work from Barth et al, 2021.
We have provided clarification within the methods pertaining to the meaning of the PC and cluster analyses (lines 350, and 367-368), to the results, regarding the functional meaning behind each cluster, and how these differ from the principal components (lines 433-435, 454-466) and reiterated these findings within the discussion (lines 555-557). Notably the patterns observed in cluster membership are consistent with prior findings (Barth et al., 2021).
3. How about the robustness across different subsamples (stroke-only, mixed clinical, controls)?
We are not sure exactly what the reviewer is asking for with this comment. Our findings show replication in another sample of people with stroke, as well as generalizability across several distinct samples of musculoskeletal, neurological and post-surgical conditions which demonstrates the robustness of this model. Specifically, figures 2 and 3 are showing the robustness of replication and generalization across the samples. If we were to only perform our analysis in people with stroke or people who served as controls, we would be unable to answer our proposed scientific question (specifically research question 2, see lines 107-113), because each sample would be missing ranges of functional severity (control constituting presumably high levels of UL function, stroke constituting presumably lower levels of UL function).
4. For the convergent validity thresholds, what R² thresholds are considered meaningful?
Thank you for this important point. Regarding the convergent validity thresholds, there are no universally accepted cut-offs for R2; rather these values should be interpreted in context. Reasonably high R2 values (e.g., 0.25-0.81) generally indicate convergent validity, but the threshold depends on the context. For instance, if the goal is to show that two measures reflect the same construct with sufficient resolution that the measures are interchangeable, then a higher R2 is required. However, in the context of this study, we encourage the reader to consider the accelerometry variables not in isolation, but as a collective set of variables that tap into the same underlying construct of interest (e.g., upper limb activity in daily life). Rather than focusing on the strength of association between each individual accelerometry variable and the PROMIS Upper Extremity measure for example, we argue that the combined pattern across variables provides an indication of convergent validity. Given the complexity and multifaceted nature of daily upper limb performance, we do not expect any single accelerometer variable to yield a high R2 on its own. Instead, consistent low-moderate relationships across multiple metrics may still provide meaningful support for construct validity. We have revised the Results section (lines 488-504) to clarify this interpretation.
5. The sample has few participants with severe UL impairment, and how might the model perform in this subgroup.
Thank you for this comment. Of the total clinical sample 20% had severe UL functional deficits which aided us in determining the lower bound of the functional range captured in our analysis. Without these individuals, the clustering analysis may have yielded fewer distinct groups (e.g., only four clusters). Individuals with severe impairments tend to cluster together given their contrasting profiles to controls on the principal components. For example, individuals in Cluster 1 (presented in blue, Figure 3) showed high preferred upper limb use, but low overall upper limb daily activity suggesting limitations in the integration of the affected limb into daily tasks and further comprising our more severely affected group. In contrast individuals in cluster 5 (presented in purple, Figure 3), demonstrated both high preferred upper limb use and high overall upper limb daily activity, indicated of more symmetrical use and greater functional performance throughout the day, and further comprising our less affected group. We have provided additional clarity for interpretation on Figure 3 (Lines 488-504).
6. How to implement the 5-variable model in clinical settings, are there any practical barriers?
These findings suggest that upper limb performance may be better represented along a continuum of functional recovery using principal components, rather than with discrete categories using clusters. In the future, a five-variable model could be implemented clinically using onboard processing in either wearable sensors or mobile devices to map an individual’s performance within a two-dimensional principal component space. This approach may offer several benefits: it can be generalized across conditions, severity levels, and avoid reliance on condition-specific clinical assessments, and reduces the need for recall of diagnostic-specific details. However, certain clinical populations may still benefit from targeted examination of specific variables, for example, the use ratio in stroke, or jerk asymmetry in ataxia, highlighting the potential need for hybrid approaches in some contexts or environments (Lines 589-599).
7. Did the authors test divergent validity with other unrelated constructs to further validate the specificity of accelerometry measures?
In this manuscript we only tested divergent validity on the Clinical Epidemiological Scale for Depression (CES-D) as an unrelated construct to performance of UL activity in daily life. We felt that if accelerometers were sensitive, they should correlate with other measures of daily life (e.g., self-report measures) which we found they did. If the accelerometers are specific, they should not correlate with unrelated measures such as depression which we found they did not. This is clarified in the Methods on lines 395-397.
Reviewer 2 Report
Comments and Suggestions for Authors
Your manuscript presents a methodologically rigorous and clinically relevant replication of a sensor-based framework for categorizing upper-limb (UL) performance, initially developed for post-stroke populations. The successful replication and extension of the five-variable, five-cluster model to diverse clinical groups—including individuals with musculoskeletal, neurological, and oncological upper-limb conditions—provides significant evidence supporting the generalizability of this approach.
Importantly, your findings highlight that while categorical cluster models are helpful, continuous metrics such as principal component scores and individual accelerometry-derived variables demonstrate stronger associations with self-reported upper-limb function. This has important implications for personalized rehabilitation tracking, as it suggests that functional performance is more accurately represented along a continuum rather than through discrete classifications.
The clarity of your methods, the thoughtful interpretation of principal component analysis, and the emphasis on convergent validity enhance the overall contribution of this work. This study marks a significant advancement in the use of wearable technology for functional monitoring and emphasizes the potential for scalable, data-driven assessments in rehabilitation practice. Thank you for your valuable contribution to the field.
Here are some suggestions to improve the article:
1. Please arrange the keywords in alphabetical order.
2. Add a sentence in the introduction that outlines how the paper is structured in the following sections.
3. The participant data is sourced from only two institutions within a single network, which may limit the generalizability of the findings across varied clinical or geographic populations. Future studies could consider validating the model across external institutions and health systems, including more diverse cultural and clinical settings, to enhance external validity.
4. The cluster-based classification approach demonstrates relatively weak associations with self-reported functional outcomes (e.g., PROMIS-9 scores), compared to principal components or individual features. Please explore alternative or enhanced clustering methods, such as supervised clustering or latent class analysis, that may better capture clinically meaningful groupings. Adding task- or context-sensitive variables also improves alignment with functional outcomes.
5. The wearable-derived variables do not differentiate between types of activities or contexts, which limits their clinical interpretability. Incorporating contextual labeling through ecological momentary assessments (EMAs), digital logs, or task recognition algorithms could enhance the model’s utility in therapy planning and behavior tracking.
6. The study does not address sensor placement accuracy, wear time adherence, or signal quality issues, which can affect data reliability in real-world settings. Please include a discussion of these limitations and, if possible, conduct a sensitivity or wear-time compliance analysis to evaluate the robustness of the findings in typical home-use conditions.
7. There is limited exploration of outliers or misalignments between predicted clusters and functional self-reports. Please provide error analysis or examples of cases where cluster assignment diverges from PROMIS outcomes. This would offer insights into model limitations and opportunities for refinement.
8. Please proofread the entire article for clarity and correctness.
Comments on the Quality of English Language
Please proofread the entire article for clarity and correctness.
Author Response
All changes to the text are in red, and line numbers correspond to the revised manuscript.
1. Please arrange the keywords in alphabetical order.
This has been corrected (lines 58-59).
2. Add a sentence in the introduction that outlines how the paper is structured in the following sections.
A statement within the introduction outlining the manuscript structure has been added (lines 107-113).
3. The participant data is sourced from only two institutions within a single network, which may limit the generalizability of the findings across varied clinical or geographic populations. Future studies could consider validating the model across external institutions and health systems, including more diverse cultural and clinical settings, to enhance external validity.
The participant data was sourced from two separate networks and includes participants from all over the United States from many different medical systems. While our data is generalizable to the healthcare and outpatient neurorehabilitation settings within the United States of America, it does not generalize to the rest of the world. We have added greater clarity for this statement in our methods section (lines 151-152).
4. The cluster-based classification approach demonstrates relatively weak associations with self-reported functional outcomes (e.g., PROMIS-9 scores), compared to principal components or individual features. Please explore alternative or enhanced clustering methods, such as supervised clustering or latent class analysis, that may better capture clinically meaningful groupings. Adding task- or context-sensitive variables also improves alignment with functional outcomes.
Although other clustering algorithms are available (e.g., mixture modeling, latent class analysis), we think the current analysis is very compelling with respect to using a continuous rather than categorical approach to understanding upper extremity activity in daily life across these different populations. For instance, mixture modeling would allow for elliptical (rather than spherical) cluster membership and latent class analysis would allow for probabilistic (rather than categorical) cluster membership. However, due to the large variation within clusters, we saw much stronger associations with either the principal components (or raw accelerometer variables) showing stronger associations with self-report measures. Additionally, although we only used the k-means clustering algorithm, we obtained the same clusters in four different samples of participants and four different sets of input variables, suggesting these clusters are robust and we would likely obtain similar clusters even if different algorithms were employed. Thus, although different clinically meaningful clusters are possible, there is strong evidence to suggest that a continuous (rather than categorical) approach to quantifying upper extremity activity will be more effective in the sampled populations.
5. The wearable-derived variables do not differentiate between types of activities or contexts, which limits their clinical interpretability. Incorporating contextual labeling through ecological momentary assessments (EMAs), digital logs, or task recognition algorithms could enhance the model’s utility in therapy planning and behavior tracking.
Yes, the reviewer is correct. The wearable-derived variables do not differentiate between types of activities, or how/when a person performs specific activities, instead, the variables are quantifying characteristics of movement (e.g. duration, magnitude, and possibly quality) as these variables are currently more clinically implementable. We note that quantification of specific, individual activities executed in daily life is an important, and complementary area under investigation by others in labs around the world. Activity-specific UL algorithms are not yet sufficiently accurate to be used in the unsupervised setting of daily life across patient populations and functional levels of severity. We have added this issue to the Discussion on lines 563-566.
6. The study does not address sensor placement accuracy, wear time adherence, or signal quality issues, which can affect data reliability in real-world settings. Please include a discussion of these limitations and, if possible, conduct a sensitivity or wear-time compliance analysis to evaluate the robustness of the findings in typical home-use conditions.
This data set contains participants who wore their devices for at least 24 hours of wear time. Within this cohort, 3.6% did not wear their devices for 24 hours and these participants were removed from accelerometry analysis. We have added weartime adherence into Table 2. Interestingly, the rate of return for usable accelerometer data (96%) was similar to the rate of return of UL PROMIS questionnaire information (96%).
Sensor placement was on bilateral wrists, so as long as the wrist straps were tight enough that they would not move or rotate on the patient’s wrists, then measurements were reliable (Lang et al., 2017). Additionally, previous research has shown that shorter durations of prescribed wear time often return greater adherence (Barak et al., 2014). Regarding signal quality, Ametris (formerly Actigraph) conforms with the requirements of the ISO 13485:2016, and is a Class II FDA Approved Medical Device (Registration Number: 3004091281) which has established reliability and validity standards of the sensors. We have added additional verbiage for clarity of the above statements within the methods section (line 212-216, 218, 221-222, and 231-233) and as a footnote for table 2.
References:
- Lang CE, Waddell KJ, Klaesner JW, Bland MD. A Method for Quantifying Upper Limb Performance in Daily Life Using Accelerometers. J Vis Exp. 2017;(122):55673. Published 2017 Apr 21. doi:10.3791/55673
- Barak S, Wu SS, Dai Y, Duncan PW, Behrman AL; Locomotor Experience Applied Post-Stroke (LEAPS) Investigative Team. Adherence to accelerometry measurement of community ambulation poststroke. Phys Ther. 2014;94(1):101-110. doi:10.2522/ptj.20120473
7. There is limited exploration of outliers or misalignments between predicted clusters and functional self-reports. Please provide error analysis or examples of cases where cluster assignment diverges from PROMIS outcomes. This would offer insights into model limitations and opportunities for refinement.
Thank you for this comment. Our findings indicate that the clusters did not align closely with the PROMIS Upper Extremity self-report measure, whereas the two principal components showed a stronger relationship. We are not recommending clustering as a model moving forward given the high within-cluster variability and limited explanatory power for PROMIS, ACS, EuroQofL, or other outcomes. In contrast, a continuous approach using PCA capture more variance. Please see table 5, and figure 4, for a comparison of the PROMIS relationship with clusters versus with the two principal components. We also would like to note a ceiling effect on the PROMIS (Kazmers et al., 2019) measure (maximum score of 62 points). This ceiling limits its sensitivity in detecting high levels of upper limb function and thus the magnitude of correspondence we can detect in relation to accelerometry measures.
- Kazmers NH, Hung M, Bounsanga J, Voss MW, Howenstein A, Tyser AR. Minimal Clinically Important Difference After Carpal Tunnel Release Using the PROMIS Platform. J Hand Surg Am. 2019;44(11):947-953.e1. doi:10.1016/j.jhsa.2019.03.006
8. Please proofread the entire article for clarity and correctness.
Thank you for your review. We have proofread the article for clarity and correctness. Of note, an error was identified in Table 4, where Akaike Information Criterion (AIC) values were incorrectly reported. We had erroneously calculated AIC values using a statistical model that contained a continuous cluster variable. We revised the model to instead contain a categorical cluster variable and updated the table accordingly. This correction does not impact the reporting of our results.
Round 2
Reviewer 1 Report
Comments and Suggestions for Authors
accept
Reviewer 2 Report
Comments and Suggestions for Authors
Thank you so much for addressing the comments in the revised version of the article.